# NDVI Characteristics and Influencing Factors of Typical Ecosystems in the Semi-Arid Region of Northern China: A Case Study of the Hulunbuir Grassland

**Yating Zhao [1,2,3], Chunming Hu [1,\*], Xi Dong [1] and Jun Li [3,4]**

[1] State Key Laboratory of Urban and Regional Ecology, Research Center for Eco-Environmental Sciences, Chinese Academy of Sciences, Beijing 100085, China
[2] School of Geographic and Environmental Sciences, Tianjin Normal University, Tianjin 300387, China
[3] Tianjin Key Laboratory of Water Resources and Environment, Tianjin Normal University, Tianjin 300387, China
[4] Academy of Eco-Civilization Development for Jing-Jin-Ji Megalopolis, Tianjin Normal University, Tianjin 300387, China
\* Correspondence: cmhu@rcees.ac.cn

**Abstract:** The semi-arid region of northern China is highly sensitive to environmental changes, especially the Hulunbuir Grassland, which has an essential ecological status and a fragile environment. This study focused on the NDVI characteristics of three different ecosystems and their dominant influencing factors. It proposed a method to show the immediate effects of factors influencing NDVI on a statistical level. The results showed that: (1) NDVI of floodplain wetland > NDVI of meadow > NDVI of sand ribbon. There were obvious differences among the three ecosystems, and the spatial distribution of NDVI was consistent with altitude. (2) The main explanatory factors were the phenological period, humidity, temperature, accumulated precipitation, runoff, and evaporation, which accounted for 68.8% of the total explanation. (3) Phenological period, humidity, and precipitation were positively correlated with NDVI. Temperature and evaporation had a positive effect on NDVI within a certain range. This study revealed the differences in environmental factors in different ecosystems, enriched the theory of NDVI influencing factors, and provided a scientific basis for future NDVI research and regional ecological conservation.

**Keywords:** NDVI; redundancy analysis; floodplain wetland; meadow; sand ribbon





## 1. Introduction

Forming the main structure of terrestrial ecosystems [1], vegetation growth status and distribution patterns are restricted by the environment [2] and can acutely reflect changes in the atmosphere, water, soil, and other components and with further feedback loops [3]. The Normalized Difference Vegetation Index (NDVI) is a quantitative value that shows the growth status of green vegetation by calculating the spectral information of ground objects. The surface information observed by remote sensing technology is combined with the visible light band and near-infrared band to calculate NDVI [4,5], scientifically reflecting the greenness, density, and growth status of vegetation. NDVI is highly sensitive to terrestrial ecosystems and global changes [6], so it is frequently used to describe ecosystem characteristics and to indicate ecological environment changes [7]. NDVI is useful to help understand the dynamic changes of ecosystems and the surface energy balance more deeply, allowing the sustainable development of regional ecological environments and social development through the systematic monitoring of long-term changes in vegetation [8,9]. It also allows a comprehensive analysis of the dynamic changes and driving factors [10–12].

Related research on the vegetation index started in the 1870s [13] but research on the driving factors of NDVI is still a hot topic [14]. There have been a large number of

studies carried out in different regions and on different scales, with different conclusions. Existing studies have demonstrated the correlation between NDVI and climate [15,16], showing that some areas are dominated by precipitation and some areas are dominated by temperature [17,18]. In their study in northern China, Lin et al. [19] found that precipitation was the dominant driving factor in temperate grassland areas and temperate desertification areas, but in the alpine vegetation area of the Qinghai–Tibet Plateau, temperature was the dominant driving factor with a positive correlation. However, in temperate desert areas, temperature had a negative correlation. Zhao et al. [20] concluded that precipitation has a great impact in arid and semi-arid regions. It is evident that there is spatial heterogeneity in NDVI and its influencing factors [3], making the study of NDVI in different ecosystems necessary.

Existing studies primarily concentrate on the annual maximum synthetic NDVI on a large scale, over the long-term, to investigate the time variation trends and responses to precipitation, temperature, and human activities [21]. The Theil–Sen estimator, Mann–Kendall method, and Hurst index [22] are the common methods used to identify year-on-year changes. Zhang et al. [23] analyzed trends in NDVI and found that both the natural environment and human activities played a role in NDVI spatial and temporal distribution. Correlational analyses and the GeoDetector model [24,25] are often used to explore the influencing factors. Yao et al. [25] analyzed the contribution to NDVI of 18 factors, including climate, soil, topography, and human activities based on the GeoDetector and concluded that the main single factors were night light brightness (51.9%), annual average air temperature (47%), and annual average atmospheric pressure (45.8%). Their results also showed that the combination of two factors had a greater impact than a single factor. For different vegetation types, Fu et al. [26] studied the vegetation of two grassland types in Inner Mongolia, finding precipitation had a significant effect on typical steppe vegetation and total nitrogen had a significant effect on meadow steppe vegetation, demonstrating the coupling mechanism of precipitation and nutrients. Recent research on NDVI has introduced several new technological approaches, such as using the Google Earth Engine (GEE) cloud platform [12] and machine learning [27,28]. He et al. [14] used GEE to quantify the effects of climate change and human activities on vegetation, revealing areas to have significant correlations with temperature (3.3%), precipitation (6.9%), and sunlight hours (20.3%). Li et al. [28] developed a machine learning model to predict monthly NDVI and found that temperature plays an important role in predicting NDVI. The studies revealed that NDVI is affected by a variety of environmental factors.

However, the existing research could be improved. First, there was a lack of high-precision, small-scale research. Most studies used 250 m MODIS images with high temporal resolution but low spatial resolution, which is not suitable for smaller areas. On the other hand, due to the long revisit cycle of high spatial resolution satellites, most of the studies used single-phase images to represent a period on a small scale, ignoring that NDVI is easily affected by many factors when making multi-year comparisons. Even within the same growing season, there will be abrupt changes due to climatic conditions, etc., so the results were conditional and uncertain [29]. Second, there was a lack of analysis of different ecosystems, instead generalizing the area and ignoring differences in the responses of different species and communities to the environment [30,31]. Third, studies usually only focused on temperature and precipitation [22], considering only a few factors [32]. However, the relationship between vegetation and climate is a complex interactive system, ignoring the role of other factors such as evaporation and insolation [25]. Moreover, the synergistic relationship and interactions between many factors have not yet been made clear. Therefore, this study focused on single images on a small scale over a long-time series, analyzing the influencing factors of commonality and individuality among three ecosystems distributed throughout the region: floodplain wetland, meadow, and sand ribbon. We aim to answer three questions: (1) What are the spatial differences in NDVI among the three typical ecosystems over a long time series? (2) Besides temperature and precipitation, are there other factors that significantly influence NDVI? What is their

explanatory power in the three ecosystems? (3) How does each driving factor differ for the three ecosystems? These questions aim to explore the differences in NDVI and the influencing factors in various ecosystems more accurately and provide theoretical support for regional ecosystem protection.

## 2. Materials and Methods

### 2.1. Study Area

The study area is located in Hulunbuir City, Inner Mongolia Autonomous Region, in the lower reaches of the Hailar River, which is a small part of the Hulunbeier Grassland. The range is 118°44′–119°42′ E and 49°1′–49°27′ N. The overall terrain is relatively flat, and the maximum elevation difference is 200 m. According to the Köppen climate classification system, the region has a temperate continental climate (DWC), with sufficient sunlight and regular periods of rain and heat. It is warm and rainy in the summer and cold and dry in the winter [33,34], with a large temperature difference within the year. The average annual temperature is about 0 degrees, and the average annual precipitation is less than 400 mm, making it a semi-arid region. The study area is dominated by herbaceous plants, the vegetation growth cycle is short, and the peak growing season is concentrated from June to September.

Three types of typical ecosystems are included in the study area, as shown in Figure 1. The first is the floodplain wetland ecosystem, which is found in the main channels of the Hailaer and Morlegher rivers. It largely unaffected by human activity and has mature development and rich communities [35]. The vegetation types are mainly reed marsh, grass-weed meadow [36], and willow thicket wetland. The second is the meadow ecosystem in the northwest of the study area, which is part of the meadow grassland Nature Reserve of Chenbarghu Qi, and is disturbed by human activities [37] such as grazing and tourism to a certain extent [38]. The vegetation types are mainly typical steppe and meadow steppe [39], and there are more communities such as *Leymus chinensis* (Trin. ex Bunge) Tzvelev, *Stipa grandis* P. A. Smirn. and *Artemisia frigida* Willd. The third is the desert ecosystem in the southwest of the study area, which is the largest sand ribbon in the Hulunbuir Sandy Land, one of the four major sandy lands in China [40]. The land is barren and sparsely populated, and the vegetation is mainly sandy annual plants, such as *Artemisia halodendron* Turcz. ex Besser [41] and weed scrub.

### 2.2. Data and Preprocessing

#### 2.2.1. Spatial Data

Over 90 Landsat remote sensing images and 2 Digital Elevation Model (DEM) images, with 30 m spatial resolution, were downloaded from the USGS official website (http://www.usgs.gov/ (accessed on 31 August 2021)). The remote sensing image time was set as April–October, showing vegetation growth from 1990 to 2020. All available images of good quality with less than 10% cloud cover were selected, and a final 70 images were screened out, ranging from 1–4 images per year. Most of them had no cloud coverage. The details of spatial data are shown in Table 1. The position and range of the three ecosystems were ground-truthed with GPS points collected during the field survey in July 2021. After preprocessing the remote sensing images, including radiometric calibration and atmospheric correction, the NDVI was calculated in batches and clipped according to the three regions. The 70 NDVI images were superimposed using the Maximum Value Composite (MVC) method to obtain the spatial distribution characteristics and this was manually reclassified into five levels referring to the natural breakpoint grading (Jenks) method to calculate the area. The subsequent results were analyzed.

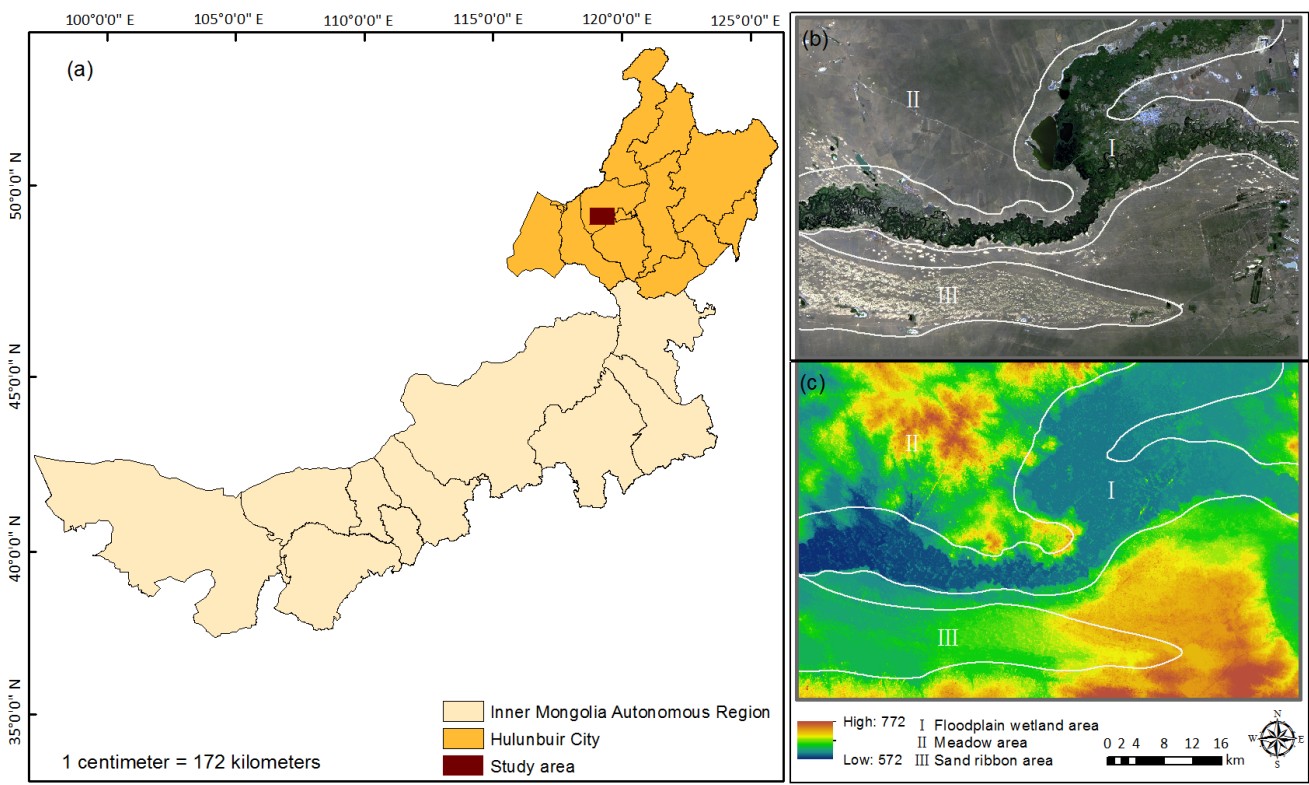

**Figure 1.** Study area. (**a**) is the location map, (**b**) is the true color remote sensing image, and (**c**) is the elevation map. (I) is the floodplain wetland area, (II) is the meadow area, and (III) is the sand ribbon area.

**Table 1.** Spatial data information.

| Data | Product\|Sensor | Track Number | Period | Count |
|---|---|---|---|---|
| DEM | ASTER GDEMV2 | 118/049 & 119/049 | 2011 | 2 |
| Landsat 5 | TM | 123/026 | 1990–2011 | 36 |
| Landsat 7 | ETM+ | 123/026 | 1999–2012 | 13 |
| Landsat 8 | OLI_TIRS | 123/026 | 2013–2020 | 21 |

### 2.2.2. Environmental Data

Daily data from the Hailar meteorological station located on the east side near the river (30 km from the center of the study area) were collected; the accumulated precipitation and maximum precipitation of the first three days, seven days and fifteen days corresponding to the acquisition data of remote sensing images were calculated, respectively. The meteorological data of the corresponding days of remote sensing images were also sorted, such as average temperature, average ground temperature, average wind speed, mean relative humidity, insolation duration, amount of evaporation, and so on. Daily runoff data from the lower Hailar River were obtained from the Hailar hydrological site. The phenological period referred to the growth height level of the main species in the study area during the growing season [42], the date divided into five phenological periods (A to E) according to the month and ten days to ensure that the differences between groups within the group pass the t-test and conform to the growth law of the main species. The germination and decay stage is phenophase A; the tillering stage is phenophase B; the flowering stage is phenophase C; the immature stage is phenophase D; and the mature stage is phenophase E. Fourteen environmental factors related to NDVI were selected, including the moisture factor, temperature factor, and time factor, as shown in Table 2.

**Table 2.** Environmental data information.

| Category | Environmental Data | Frequency | Units | Source |
|---|---|---|---|---|
| Moisture | Accumulated precipitation in the first three days | 3-Day | mm | Hailar meteorological station |
| | The maximum precipitation in the first three days | | | |
| | Accumulated precipitation in the first seven days | 7-Day | | |
| | The maximum precipitation in the first seven days | | | |
| | Accumulated precipitation in the first fifteen days | 15-Day | | |
| | The maximum precipitation in the first fifteen days | | | |
| | Mean relative humidity | Daily | % | |
| | Runoff | Daily | m$^3$ | Hailar hydrological station |
| Temperature | Average temperature | Daily | °C | Hailar meteorological station |
| | Average ground temperature | | | |
| | Insolation duration | | h | |
| | Amount of evaporation | | mm | |
| | Average wind speed | | m/s | |
| Time | Phenological period | 10-Day | - | - |

*2.3. Research Method*

This study focused on NDVI, exploring the driving factors and distinguishing the differences between ecosystems through statistical analysis of multi-phase data. In order to reduce the disturbance caused by a difference in environmental factors observed at different times, we selected 70 remote sensing images over 30 years for this research. The main technical methods used were geographic information analysis, with the help of ENVI and ArcGIS software. Statistical data analysis was carried out using Canoco and SPSS software. The process was as follows: (1) calculate the NDVI of each ecosystem and the mean value of pixels greater than zero in each period; (2) count the NDVI data and compose the spatial distribution map; (3) analyze the correlation of environmental factors; (4) explore the differences of different ecosystems; (5) further obtain the quantitative description of the contribution of dominant factors through redundancy analysis; (6) group the main driving factors separately to explore the mechanism of impact on each ecosystem. More details are shown in Figure 2.

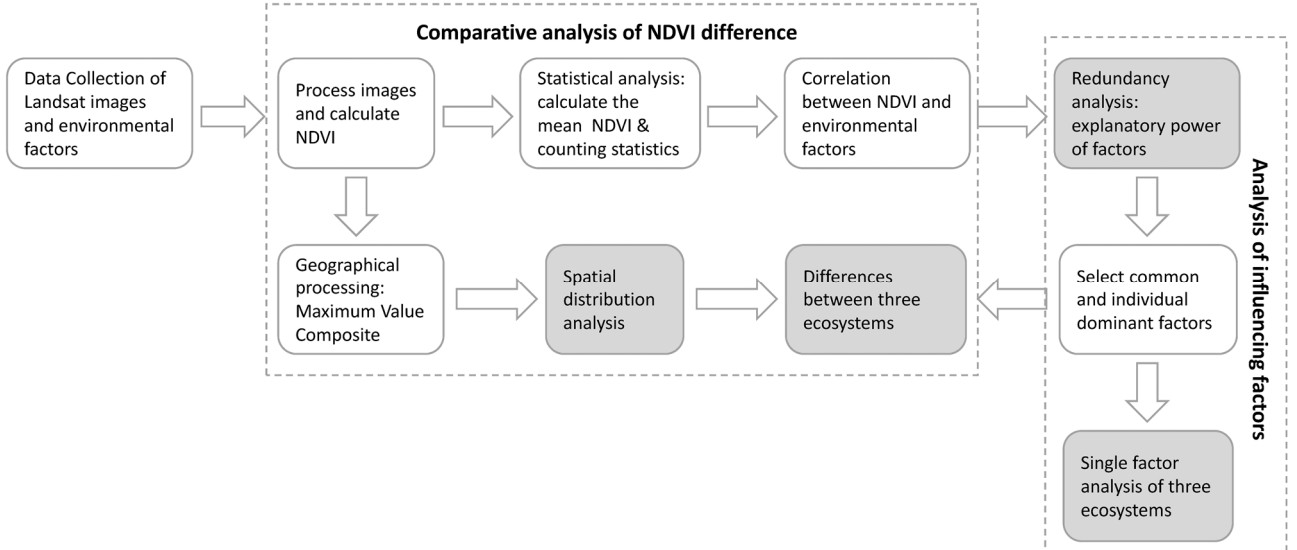

**Figure 2.** The workflow of the research.

1. Normalized Difference Vegetation Index

NDVI is a quantitative value that shows the growth status of green vegetation calculated from the spectral information of ground objects [43]. The formula is:

$$\text{NDVI} = \frac{NIR - R}{NIR + R} \tag{1}$$

where, *R* is the red band, and *NIR* is the near infrared band, which corresponds to the third and fourth bands in Landsat4/5 TM and Landsat7 ETM+, and the fourth and fifth bands in Landsat 8 OLI, respectively.

2. Correlation analysis

The correlation coefficient is obtained by correlation analysis of the factors. Its calculation formula [44] is:

$$r_{xy} = \frac{\sum_{i=1}^{n}(x_i - \bar{x})(y_i - \bar{y})}{\sqrt{\sum_{i=1}^{n}(x_i - \bar{x})^2 \sum_{i=1}^{n}(y_i - \bar{y})^2}} \tag{2}$$

where, $r_{xy}$ is the correlation coefficient between variables $x$ and $y$, and the value range from $-1$ to 1. Positive values represent a positive correlation between variables; negative values represent a negative correlation between variables; the larger the absolute value of $r$, the stronger the correlation; $n = 70$ is the sample size, $i = 1, 2, 3, \ldots, 70$.

3. Redundancy analysis

Redundancy analysis (RDA) is a ranking method combining multivariate regression analysis and Principal Component Analysis (PCA) [45]. It is implemented by Canoco, a multivariate statistical analysis software for ecological data that can explicitly explore the relationship between response variables and explanatory variables. The scores of correlation variation between them are visually displayed by the double order diagram [46]. In this study, the response variable is NDVI, and the explanatory variable is the dominant driving factor, so the correlation and contribution scores of each explanatory variable and response variable can be quantitatively analyzed.

$$Y_{ik} = b_{1k} \times c_{11} \times Z_{1i} + b_{1k} \times c_{12} \times Z_{2i} + \ldots + b_{2k} \times c_{21} \times Z_{3i} \tag{3}$$

As in the above formula, the left side of the equation is the response variable Y, the right side is the explanatory variable Z, *b* and *c* are the estimated coefficients of a variable, and $b \times c$ represents the parameters of the multiple regression model [47].

## 3. Results

### 3.1. Statistical Characteristics of NDVI in Each Typical Ecosystem

The statistical characteristics of the NDVI data of floodplain wetland, meadow, and sand ribbon in each period are shown in Figure 3. The proportion of low values in the three regions was large, and the mean value was low. On the whole, the NDVI of the floodplain wetland was higher than that of the meadow and the sand ribbon ecosystems. The floodplain wetland data showed an approximately normal distribution, with a large range of 0–0.6; the mean and median were close, the center of gravity was low, and the extreme value tended to be an outlier. The meadow and sand ribbon data showed a skewed distribution with a low mean and center of gravity, and the majority of the values were concentrated in the interval less than 0.1. There was a significant difference between the mean and median, and the extreme value also had a tendency to be an outlier. Among them, the value range of meadow ecosystems was wider than that of sand ribbon ecosystems, while the distribution was more unbalanced and the value was slightly higher than that of sand ribbon.

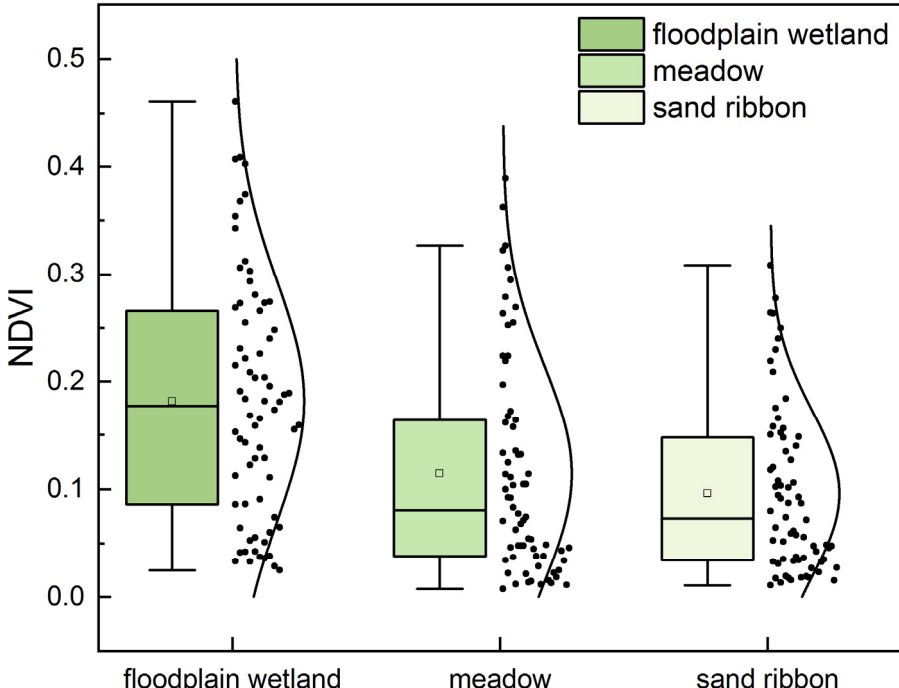

**Figure 3.** Statistical plots of NDVI for each ecosystem. The horizontal line in the boxplot is the median, and the hollow point is the mean. The black point on the right is each of the 70 images' mean value of NDVI, and the curve is the distribution curve.

The statistical results of the floodplain wetland were explored in detail and 70 images with an uneven time distribution were selected; statistical results in line with normal distribution were obtained, while other ecosystems in the same period were not the same. Compared with the other two ecosystems, the vegetation growth conditions of the floodplain wetland can be satisfied in most periods. However, the other two systems are vulnerable to the stress of environmental conditions or human activities, with more data clustered at low values and an irregular distribution.

### 3.2. Spatial Distribution of NDVI in Each Typical Ecosystem

The spatial distribution characteristics are shown in Figure 4. There were obvious differences between the three regions. The NDVI values of the floodplain wetland in the middle of the region were mainly greater than 0.5, which were concentrated near the river. It was also the area with the lowest elevation, where the vegetation coverage was the highest and the growth was the best. The NDVI values of the northern meadow were dominated by values in the range of 0.3–0.5, especially 0.4–0.5, which accounted for 56.6%, and the growth trend gradually improved from west to east. This was related to the altitude, where the higher the altitude, the higher the NDVI value; this trend was the opposite of that in the floodplain wetland. The southern sand ribbon consisted of many small independent sand masses forming a continuous sandy land with low NDVI values. A large area with values less than 0.3 was clearly visible on the map, where vegetation was not growing well. The NDVI value around the sand mass was relatively high and it had the same high eastern and low western trend as the meadow.

By comparing the three groups, obvious differences were found in the NDVI values of different ecosystems in one region, which were related to the type, abundance, and growth of vegetation. The NDVI of the floodplain wetland near the water source was significantly higher than that of the other two regions, and only a small area of NDVI was higher than 0.5 in the meadow and sand ribbon, which confirmed the rationality of regional division and led to further exploration of the reasons for these differences.

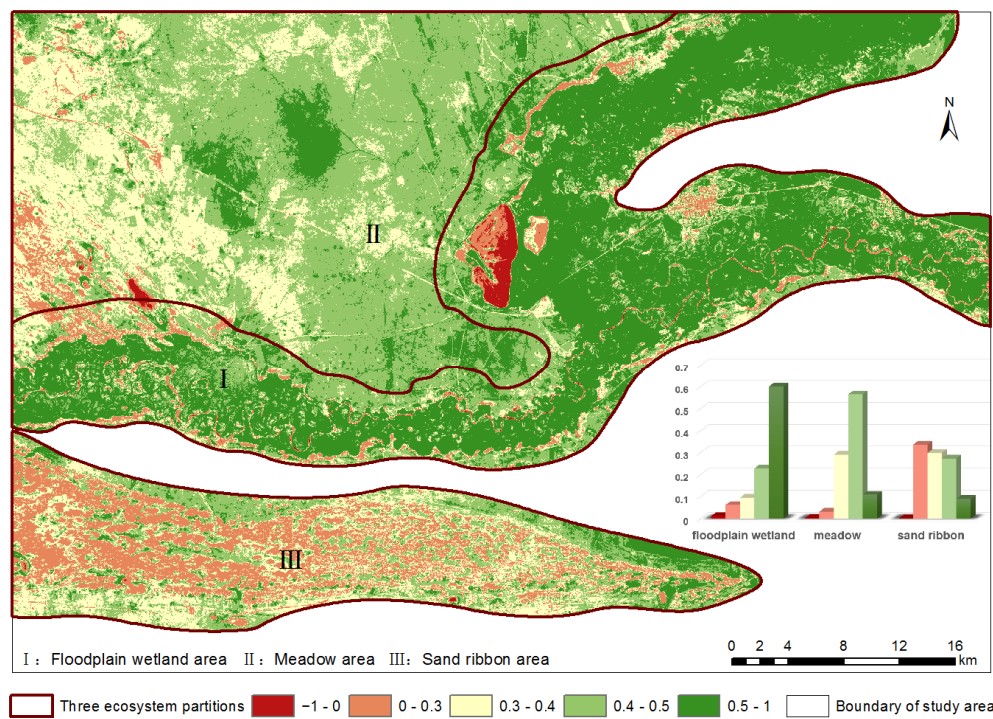

I ： Floodplain wetland area   II ： Meadow area   III ： Sand ribbon area

| | Three ecosystem partitions | | −1 - 0 | | 0 - 0.3 | | 0.3 - 0.4 | | 0.4 - 0.5 | | 0.5 - 1 | | Boundary of study area |

**Figure 4.** Spatial distribution of the NDVI maximum value composite and the area ratio histogram.

*3.3. Dominant Driving Factors of NDVI*

3.3.1. Correlation Analysis of Driving Factors

As shown in Table 3, the correlations between the NDVI of different ecosystems and 14 environmental factors were analyzed. It can be seen that the correlation between most factors and NDVI was significant at the level of 0.05, and phenology had the highest correlation. Among precipitation factors, the accumulated precipitation had a higher correlation than the maximum precipitation, especially the accumulated precipitation in the first seven days, which had the greatest correlation with the three systems' NDVI, especially the correlation coefficient with sand ribbon. In addition, the response of sand ribbon to precipitation in the first three days, significant at the level of 0.01, was higher than that of wetland and meadow. Average temperature had a greater impact on ecosystem NDVI than average ground temperature, especially for floodplain wetlands, and the correlation coefficient exceeded 0.5, indicating that floodplain wetlands had a greater vegetation response to temperature. It was also more significantly correlated with insolation duration and average wind speed than other systems. The correlation with runoff was higher and had a significant level of 0.05 for the meadow compared with other systems.

Through correlation analysis, it can be seen that the factors significantly related to the commonality of the three systems are precipitation in the first seven and fifteen days, phenology, temperature, and relative humidity. There were different individual factors in each system. Insolation duration and average wind speed were only significantly correlated with floodplain wetland. Runoff was only significantly correlated with meadow. Personality factors were closely related to the characteristics of each ecosystem.

As shown in Table 3, the phenological period, humidity, and precipitation factors of the three ecosystems were significantly correlated with NDVI at the level of 0.01, but only the phenological period had a high correlation coefficient, while other factors had a moderate correlation. The reason may be that the data in the study was from April to October, with a long time span and large changes in NDVI and environmental factors. Therefore, we analyzed the correlation of NDVI and other environmental factors in five phenological periods and selected the factors significantly correlated with NDVI in Table 4. From Phenophase A to E, the vegetation grew progressively better and higher. There is no

list for Phenophase C because the correlation is not significant. The correlation coefficients also improved. It can be seen that there are obvious differences in the correlation between NDVI and various environmental factors in different phenological periods. For example, Phenophase A, when vegetation height is the lowest, has a very high correlation coefficient with precipitation in the first three days and also with evaporation; while Phenophase B, when vegetation is growing, has a higher correlation with insolation duration and ground temperature. Phenophase E shows the best growth in the period with the highest temperature. At this time, the correlation between NDVI and humidity and runoff increases, which means the demand for water is increasing. In general, the significant correlation between the dominant factors mentioned above and NDVI are valid.

**Table 3.** Correlation analysis between driving factors and the NDVI of each ecosystem.

| Driving Factors | Floodplain Wetland | Meadow | Sand Ribbon |
|---|---|---|---|
| Accumulated precipitation in the first three days | 0.239 * | 0.316 ** | 0.348 ** |
| The maximum precipitation in the first three days | 0.227 | 0.305 * | 0.328 ** |
| Accumulated precipitation in the first seven days | 0.468 ** | 0.478 ** | 0.483 ** |
| The maximum precipitation in the first seven days | 0.405 ** | 0.424 ** | 0.432 ** |
| Accumulated precipitation in the first fifteen days | 0.452 ** | 0.445 ** | 0.460 ** |
| The maximum precipitation in the first fifteen days | 0.324 ** | 0.299 * | 0.322 ** |
| Phenological period | 0.758 ** | 0.608 ** | 0.639 ** |
| Runoff | 0.122 | 0.246 * | 0.202 |
| Average temperature | 0.540 ** | 0.315 ** | 0.295 * |
| Insolation duration | 0.249 * | 0.127 | 0.090 |
| Average ground temperature | 0.526 ** | 0.294 * | 0.260 * |
| Mean relative humidity | 0.409 ** | 0.524 ** | 0.529 ** |
| Amount of evaporation | 0.115 | 0.004 | −0.036 |
| Average wind speed | −0.260 * | −0.203 | −0.192 |

\* and \*\* represent $p < 0.05$ and $p < 0.01$, respectively.

**Table 4.** Correlation analysis of different phenological periods.

| | | The Most Important Factor and Its Correlation | | | The Second Important Factor and Its Correlation | | |
|---|---|---|---|---|---|---|---|
| | | Factor | Pearson Correlation | *p* | Factor | Pearson Correlation | *p* |
| **A Phenophase** | Floodplain wetland | Accumulated precipitation in the first 3 days | 0.832 * | 0.010 | Amount of evaporation | 0.535 | 0.172 |
| | Meadow | | 0.953 ** | 0.000 | | 0.723 * | 0.043 |
| | Sand ribbon | | 0.951 ** | 0.000 | | 0.795 * | 0.018 |
| **B Phenophase** | Floodplain wetland | Average ground temperature | 0.604 * | 0.017 | Insolation duration | 0.847 ** | 0.000 |
| | Meadow | | 0.533 * | 0.041 | | 0.740 ** | 0.002 |
| | Sand ribbon | | 0.458 | 0.086 | | 0.683 ** | 0.005 |
| **D Phenophase** | Floodplain wetland | Accumulated precipitation in the first 7 days | 0.652 ** | 0.008 | Mean relative humidity | 0.434 | 0.106 |
| | Meadow | | 0.623 * | 0.013 | | 0.631 * | 0.012 |
| | Sand ribbon | | 0.635 * | 0.011 | | 0.541 * | 0.037 |
| **E Phenophase** | Floodplain wetland | Mean relative humidity | 0.746 ** | 0.000 | Runoff | 0.573 * | 0.010 |
| | Meadow | | 0.792 ** | 0.000 | | 0.706 ** | 0.001 |
| | Sand ribbon | | 0.802 ** | 0.000 | | 0.573 * | 0.010 |

\* and \*\* represent $p < 0.05$ and $p < 0.01$, respectively.

3.3.2. RDA for Driving Factors

In order to explore the relationship between factors further, the RDA analysis was carried out in the ecology software Canoco 5. This method is not limited to a linear analysis of the relationship between independent variables and dependent variables, instead, it uses a multidimensional gradient to analyze the regression relationship between response variables and explanatory variables. This method has been widely used in community research and has now been extended to more fields [46,47].

As shown in Figure 5, the blue arrow represents the response variable, which is the fitting of the NDVI values of the three ecosystems. It can be seen that the distribution of meadow and sand ribbon was similar, with some difference between them and the floodplain wetland ecosystems. The red arrows indicate the 14 explanatory variables; all factors except wind speed pointed in the direction of increasing values on the first sorting axis, because these factors are positively correlated with NDVI. The moisture factors were located in the first quadrant. The temperature factors pointed in the negative direction of the second sorting axis. The low correlation between the two datasets indicated large differences in the impact mechanisms on the ecosystem. The longest projection on the first sorting axis was the phenological period, indicating that its overall contribution to the factor variance was the highest. The shortest was evaporation. The relationship between NDVI and the driving factors clearly showed that the highest NDVI explanation rate for all three ecosystems was for the phenological period, with floodplain wetland being more influenced by temperature-related factors, and meadow and sand ribbon being more influenced by moisture-related factors, probably because floodplain wetlands were distributed on both sides of the river and had good moisture conditions. In addition to the phenological period, floodplain wetlands were most affected by average temperature but least correlated with runoff; meadows and sand ribbons were most affected by mean relative humidity and least correlated with evaporation. Based on the results of the RDA analysis, it can be concluded that moisture condition is the most important driving factor for ecosystems in semi-arid areas and that its influence on plant growth is higher than that of other environmental factors such as temperature and insolation duration.

It can be seen from the biordered graph obtained by RDA that there is also a strong correlation between each driving factor. In order to better identify the degree of influence of each factor on NDVI, forward selection is used to eliminate the collinearity between factors, and the top six factors with synergistic explanatory power for the three ecosystems from high to low at the significant level are screened out. As shown in Figure 6, the total explanation reached 68.8% when the phenological period, mean relative humidity, average temperature, accumulated precipitation in the first seven days, runoff, and amount of evaporation were all taken into account. With the exception of phenology, the explanation of other factors was slightly lower. The factor explanation of floodplain wetland was the highest, reaching 77.8%, and phenological period occupied an overwhelming proportion, which was obviously different from other ecosystems. Insolation duration and average ground temperature also had significant effects, and the 1% contribution of runoff is not significant. The NDVI of meadow and sand ribbon ecosystems had a relatively similar composition of environmental explanations, and the difference was reflected in the higher explanatory power of phenological period for sand ribbon and the higher explanatory power of the amount of evaporation and runoff for meadow.

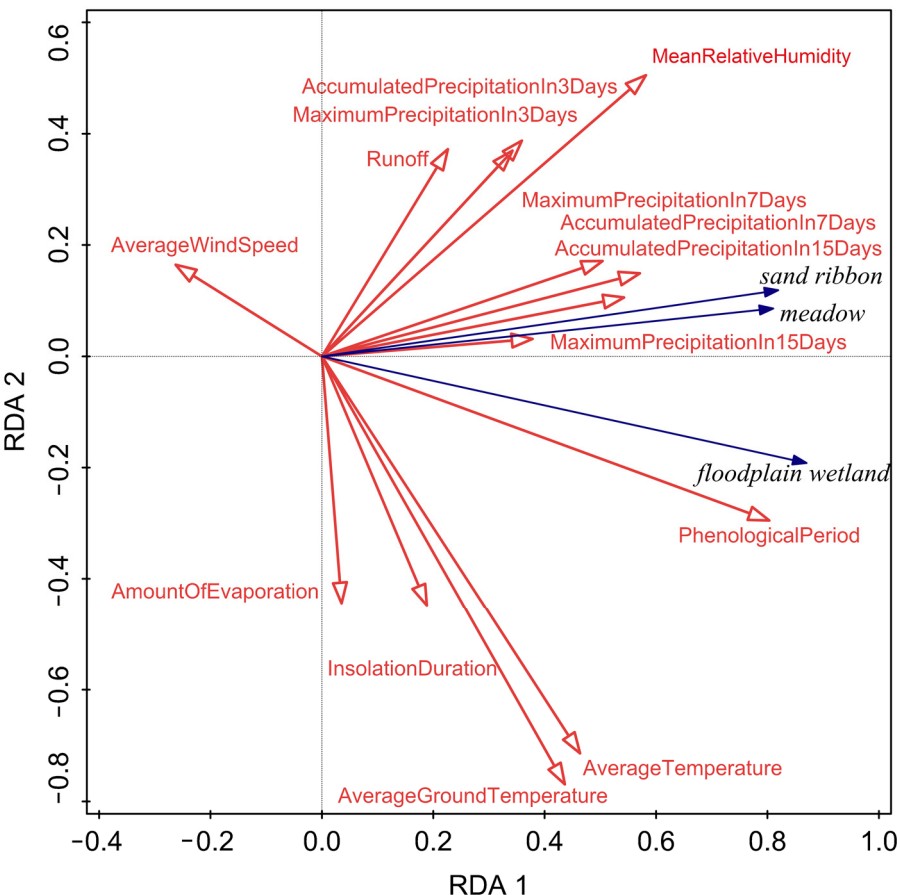

**Figure 5.** Biordered graph of RDA. RDA 1 represents the fitting value. RDA 2 represents the regression residual. The direction of the arrow indicates the value increases; its length is proportional to the maximum rate of change. The angle between arrows reflects the correlation. The acute, obtuse, and vertical angles represent positive, negative, and non-correlation, respectively.

### 3.4. The Influence of the Dominant Driving Factors on Each Typical Ecosystem's NDVI

Each of the six dominant driving factors was divided into five equally spaced groups, and the characteristics and effects on each ecosystem were observed. The main figures are the RDA triorder diagrams displayed in groups, and the secondary figures are the NDVI statistical diagrams within each group of the three systems. The points in the triorder diagram are 70 sample points with full factor attributes, and the coordinates are the scores of the samples on the two axes. The distance between the points indicates the dissimilarity between each sample. The projection of the sample points along the arrow of the environmental variable is approximately the size of the data value of the factor, where the value close to the origin is the average value, and the value increases along the direction of the arrow. Therefore, the sample point group after grouping the data has the distribution characteristics of an increasing gradient in a direction perpendicular to the arrow. This distribution trend is more obvious in the triorder graph.

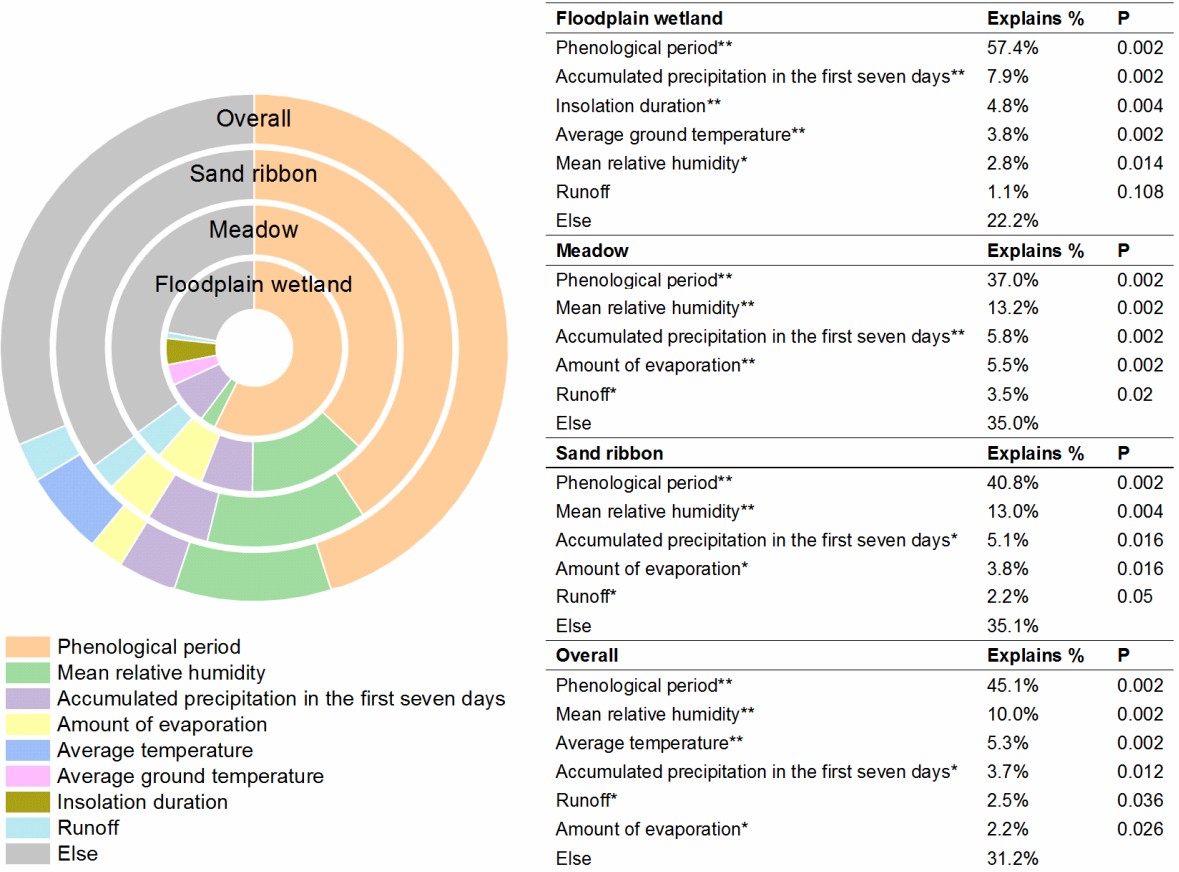

| Floodplain wetland | Explains % | P |
|---|---|---|
| Phenological period** | 57.4% | 0.002 |
| Accumulated precipitation in the first seven days** | 7.9% | 0.002 |
| Insolation duration** | 4.8% | 0.004 |
| Average ground temperature** | 3.8% | 0.002 |
| Mean relative humidity* | 2.8% | 0.014 |
| Runoff | 1.1% | 0.108 |
| Else | 22.2% | |
| **Meadow** | **Explains %** | **P** |
| Phenological period** | 37.0% | 0.002 |
| Mean relative humidity** | 13.2% | 0.002 |
| Accumulated precipitation in the first seven days** | 5.8% | 0.002 |
| Amount of evaporation** | 5.5% | 0.002 |
| Runoff* | 3.5% | 0.02 |
| Else | 35.0% | |
| **Sand ribbon** | **Explains %** | **P** |
| Phenological period** | 40.8% | 0.002 |
| Mean relative humidity** | 13.0% | 0.004 |
| Accumulated precipitation in the first seven days* | 5.1% | 0.016 |
| Amount of evaporation* | 3.8% | 0.016 |
| Runoff* | 2.2% | 0.05 |
| Else | 35.1% | |
| **Overall** | **Explains %** | **P** |
| Phenological period** | 45.1% | 0.002 |
| Mean relative humidity** | 10.0% | 0.002 |
| Average temperature** | 5.3% | 0.002 |
| Accumulated precipitation in the first seven days* | 3.7% | 0.012 |
| Runoff* | 2.5% | 0.036 |
| Amount of evaporation* | 2.2% | 0.026 |
| Else | 31.2% | |

**Figure 6.** Pie chart of NDVI explanatory power. * and ** represent $p < 0.05$ and $p < 0.01$, respectively.

It can be seen from Figure 7 that the gradient distribution trend of sample points for the two factors of phonological period and humidity was obvious, with some positive correlation between the corresponding and NDVI values, especially between phonological period and floodplain wetland, and between humidity and meadow and sand ribbon. The grouped statistics of precipitation factors had the characteristic that NDVI values increased more with the increase of precipitation, but due to the low frequency of large precipitation, the statistical samples were few, which made them sporadic and mutable. The evaporation factor had the characteristic of a single peak curve. When the evaporation was low, the NDVI value increased with the increase in evaporation. However, after reaching a peak value, the increase in evaporation led to a decrease in NDVI value. In the first group of temperature factor, the temperature was lower than 0°, which is not suitable for vegetation growth, so the NDVI value was abnormally low. Then, the NDVI value tended to increase with the increase in temperature. However, when the T4 group was 15–20°, the NDVI value first decreased, and then rose in the T5 group. Although the temperature was suitable, the humidity was low, resulting in low NDVI values. There was no obvious rule in the runoff factor characteristics.

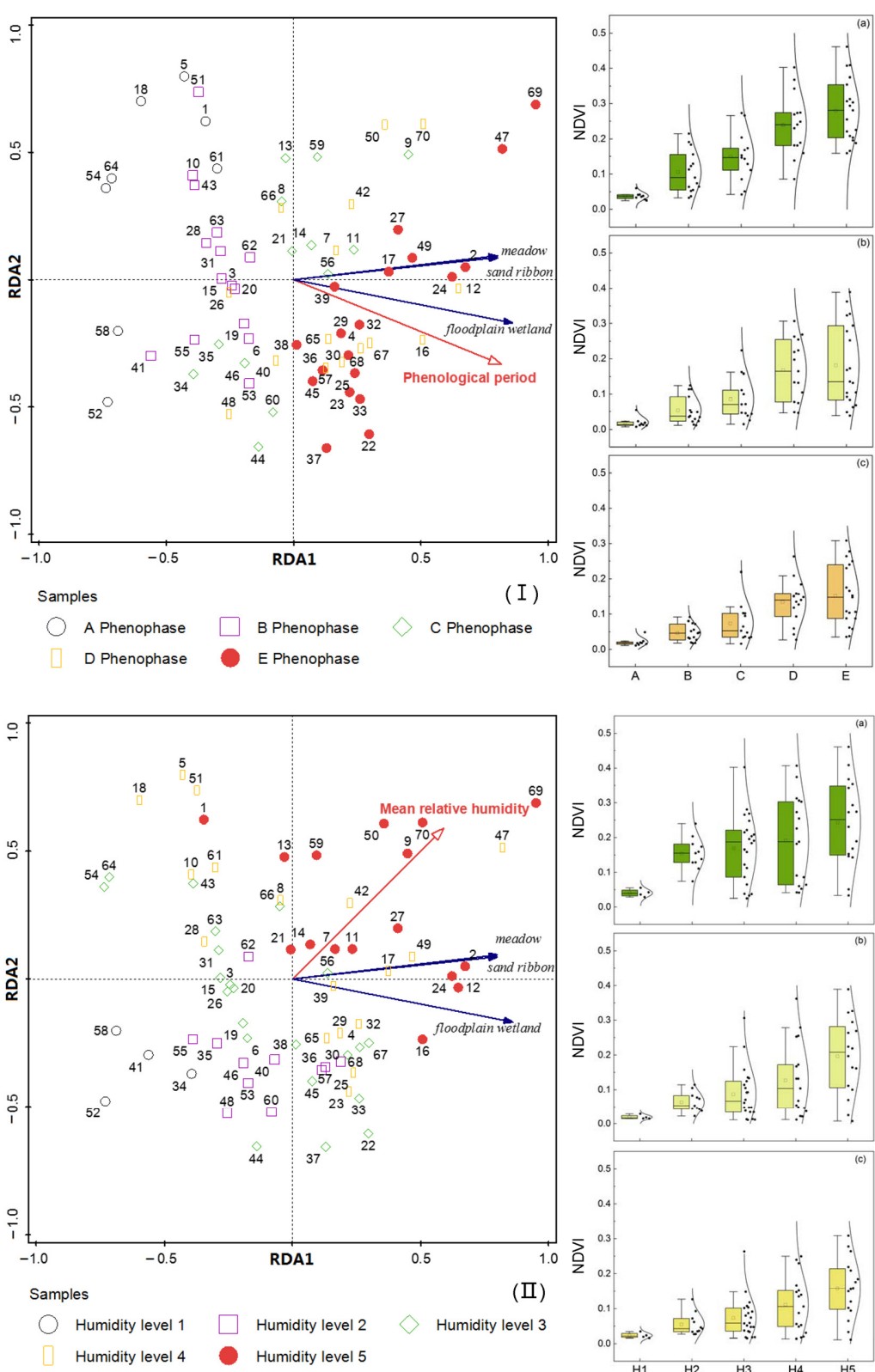

**Figure 7.** *Cont.*

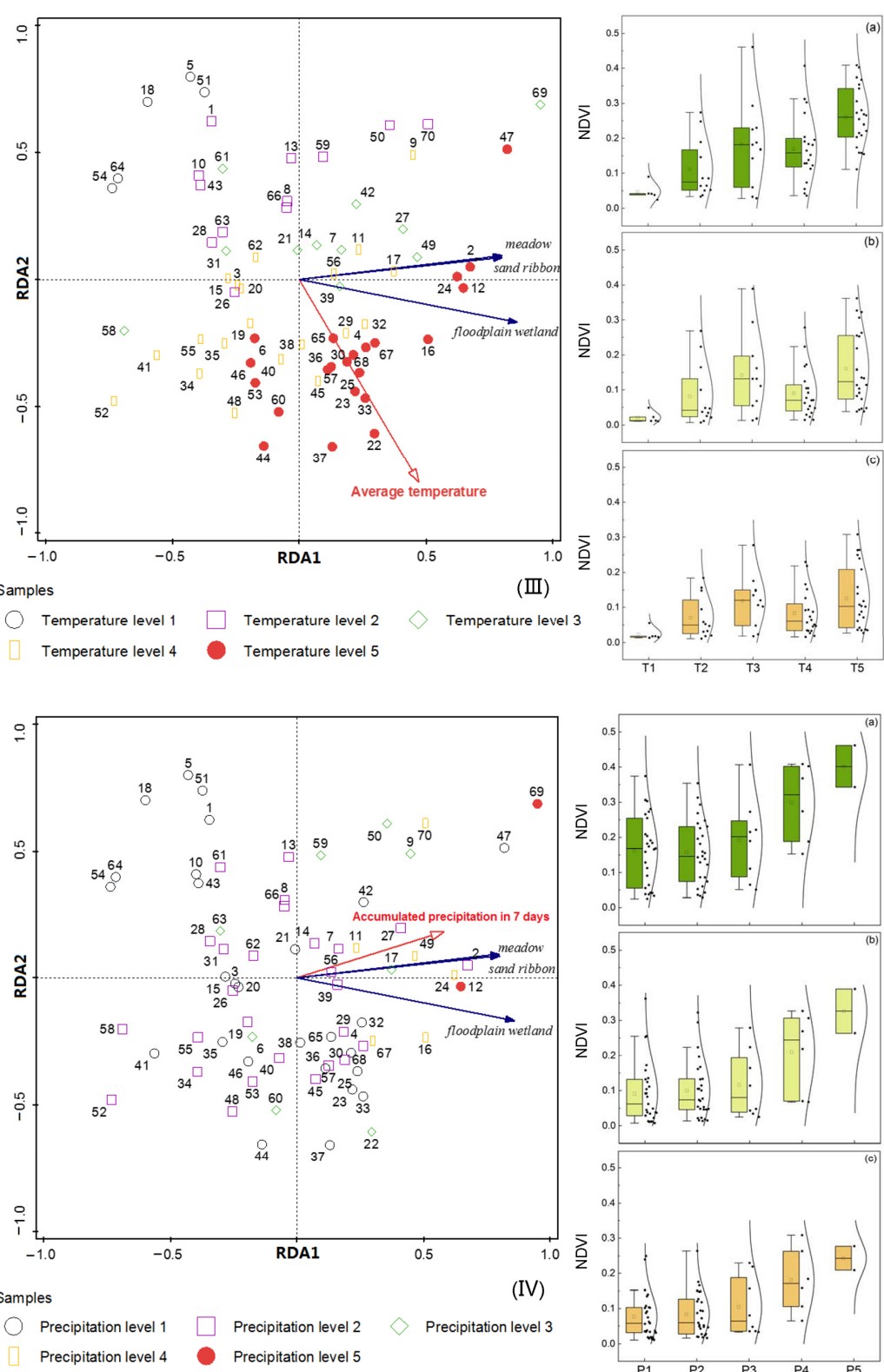

**Figure 7.** *Cont.*

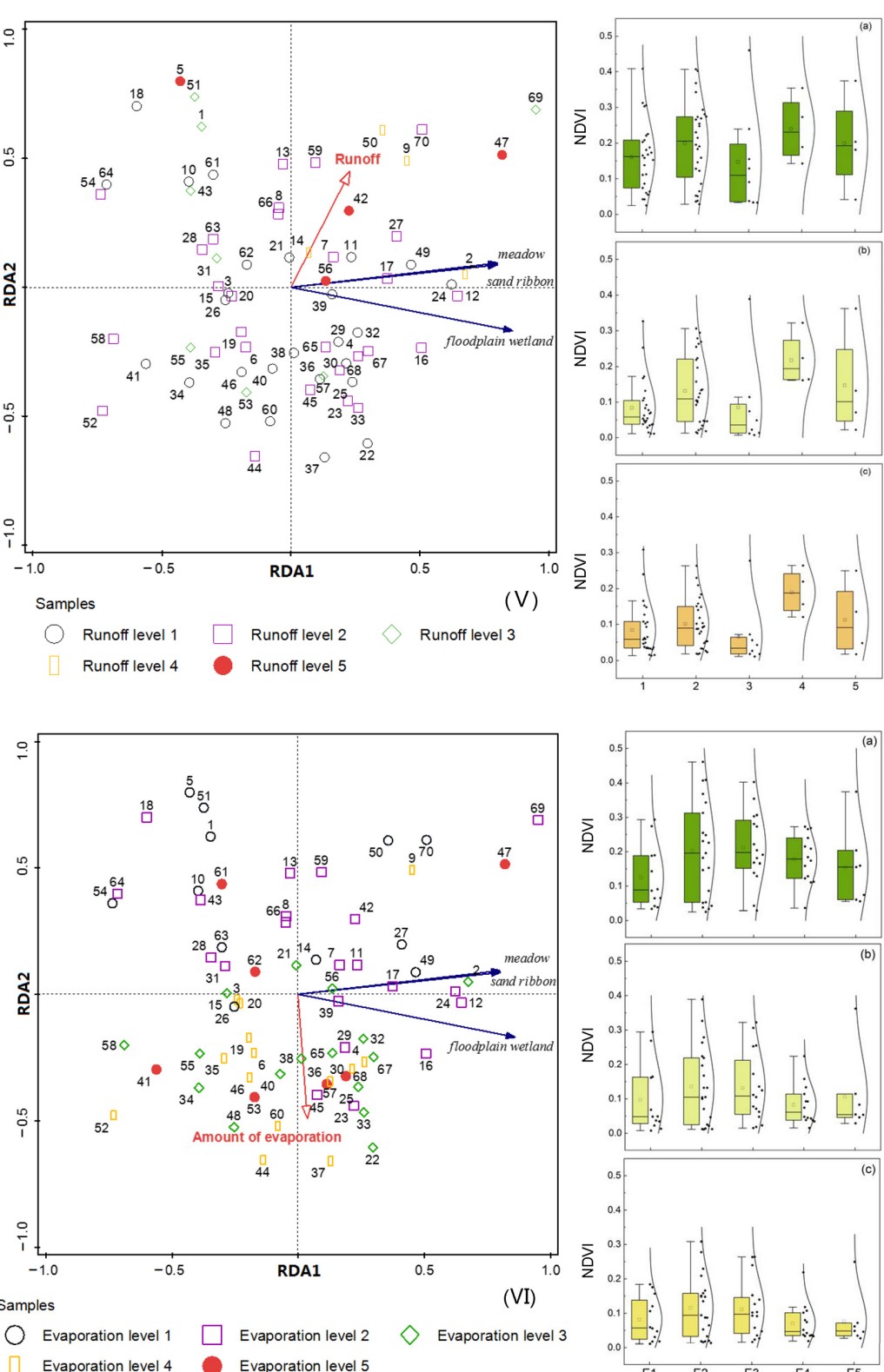

**Figure 7.** Driving factors grouping characteristics and data statistical graph. (**I**) is the phenological period characteristic, (**II**) is the humidity characteristic, (**III**) is the temperature characteristic, (**IV**) is the precipitation characteristic, (**V**) is the runoff characteristic, and (**VI**) is the evaporation characteristic. (**a**) is the statistical distribution of floodplain wetland, (**b**) is the statistical distribution of meadow, and (**c**) is the statistical distribution of sand ribbon. The horizontal line in the boxplot is the median, and the hollow point is the mean. The black point on the right is the NDVI, and the curve is the distribution curve.



## 4. Discussion

### 4.1. Differences among Three Ecosystems

The study area is a special region located in the vast Hulunbuir Grassland. The three typical ecosystems are in the same climate conditions and unaffected by human activities. Therefore, it is valuable to explore the performance of the NDVI of the three ecosystems and their responses to the environment under the same natural factors. Gu et al. [31] found a maximum value of 250 m NDVI from 1981 to 2019 in Hulun Lake Basin, their NDVI results being similar to the spatial MVC results in this study.

With certain natural conditions, the floodplain wetland is one of the most biodiverse ecosystems in the high-latitude semi-arid area, where vegetation is the bridge between aquatic ecosystem and terrestrial ecosystems [48]. The vegetation growth of the Hailar River floodplain wetland conforms to the law of nature. The NDVI was highly correlated with phenological period, and the vegetation growth is better in the low-lying area closer to the river [49]. There has been relatively little research on the Hailar River floodplain wetland until now [35,49]. Feng et al. [50] found the mean NDVI of Panjin Reed Wetland in the same temperate zone was 0.6, but in July, when the growth peak was reached, the NDVI could reach more than 0.7. The result confirmed the highly consistent relationship between NDVI and phenological period in this paper. The main species of this wetland is *Phragmites australis* (Cav.) Trin. ex Steud., so the NDVI is higher than that of the floodplain wetland. Compared with the complex ecological environment of the floodplain wetland, the morphology of this wetland does not easily change, with little variation within the year. Yan et al. [17] studied the wetland vegetation in China, finding samples with an NDVI value lower than 0.3 were concentrated in semi-arid and arid areas. In areas where there is enough water for the growth of wetland vegetation, temperature rises will promote growth, which is consistent with the results of this study.

The NDVI range of the meadow had a large span and the variance between the single-phase mean value is large. The meadow is easily affected by the environment and periodic human interference such as grazing and weeding [20]. Piao et al. [51] studied the temperate grassland in northern China on a large scale. From 1982 to 1998, the mean NDVI of the steppe corresponding to the location of our study area was between 0.26 and 0.31. In addition, the correlation between mean growing season NDVI and precipitation was higher than that with temperature, which is also similar to the results of this study.

The sand ribbon ecosystem is in the Hulunbuir Sandy Land, the fourth largest sandy land in China. Due to deforestation, human destruction, and other reasons, the degradation of grassland began in the middle of the last century [33]. The NDVI here was mainly less than 0.3, and the most of single-phase mean values were under 0.1, similar to the NDVI of areas of the same latitude in Eurasia from Andela et al.'s study [32]. Tian et al. [52] studied the vegetation of 10 deserts and sandlands in Inner Mongolia, finding that the mean NDVI of Hulunbuir Sandland was higher than that of Ujimqin Sandland, Mu Us Sandland, and Hunshadake Sandland. This was probably due to the better moisture condition of Hulunbuir Sandland. The study also showed it had more anomalous degradation compared with other sandlands.

This study area is located in the semi-arid zone of a temperate climate. Meadow grassland is the most widely distributed ecosystem type, and the floodplain wetland was developed in the channel area with sufficient moisture conditions, while grassland was degraded to sandy land in the area disturbed by human activities [22]. The difference between NDVI values in the three ecosystems is very obvious: the performance is not the same in different periods, and neither is the response to the environment, indicating that the driving factors of NDVI for different ecosystems are different even under the same climatic conditions.

### 4.2. Influence of Dominant Driving Factors

The purpose of introducing phenology as a factor was not to explore the influence of phenological period on vegetation, but to study the statistical significance of phenological

period and NDVI value. The value of NDVI is highly correlated with the vegetation growth rhythm and phenological period, and vegetation growth height and greenness are the embodiments of NDVI. However, their phenological period stages are different for different ecosystems or species [53]; therefore, comparing NDVI values of different phenological period stages is not convincing. In this study area, the vegetation of floodplain wetland mainly grew naturally [53], with sufficient water and soil nutrients, diverse plant species, and good growth. It was less affected by human activity and follows natural rhythms, so it had a higher correlation with the phenological period. Although there were similar characteristics between meadow and sand ribbon, meadow ecosystems were interfered with by human social activities such as grazing [37], and in mid-September, large-scale mowing led to an abnormal decrease in NDVI [54], which did not occur in the other two ecosystems. Restricted by the soil and water environment, the vegetation in sandy land was sparse and had poor growth. Although there was no interference from human activities, it was greatly affected by climate. Therefore, the explanatory power of phenological period in the three systems reflected the power of the effect of environmental stress on the ecosystem; that is, the external environment had the least influence on vegetation in the floodplain wetland, followed by sand ribbon, restricted by significant climate factors such as precipitation. Finally, the meadow ecosystem was most influenced, being stressed by significant environmental factors and human activities, resulting in growth conditions that cannot be satisfied in some periods.

The Hulunbuir Grassland in the semi-arid region was more affected by moisture conditions such as humidity and precipitation than other climatic factors such as temperature [31]. Sand ribbon ecosystem with low soil moisture content had a stronger immediate response to precipitation compared to other areas. In contrast, the floodplain wetland, with its sufficient groundwater supply, had a low demand for humidity and a higher response to temperature than the meadow and sand ribbon ecosystems. This result was also consistent with the findings of Hou Guanglei [55], Yang Da [56], and other scholars. On the other hand, compared with precipitation, humidity had a more direct effect on the available water for vegetation [57] and better binomial fitting with each system as shown in Figure 8. The overall performance showed that NDVI increased with an increase in humidity, but with apparent differences among the three systems; the meadow ecosystem stood out, showing a more drastic increase with higher humidity.

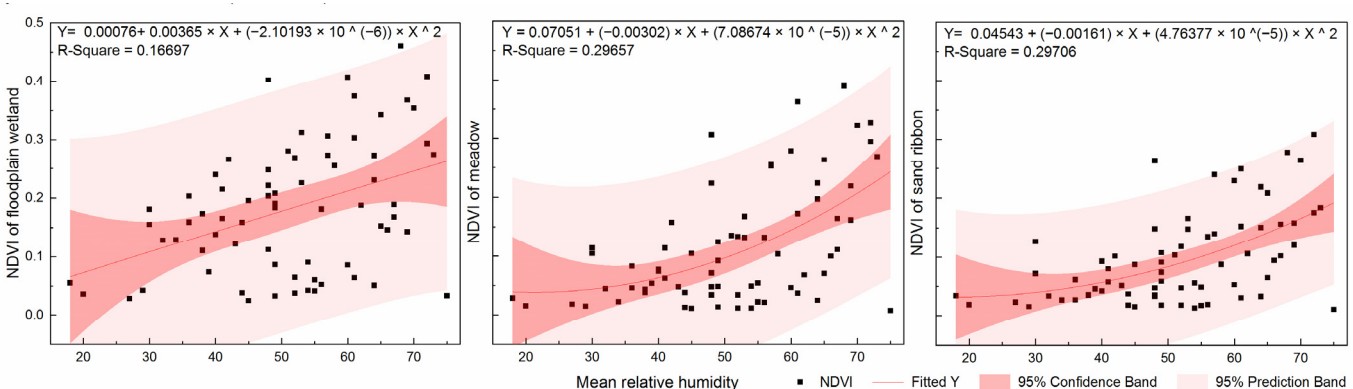

**Figure 8.** NDVI-humidity scatter fitting curve.

Because the study area is located in the lower reaches of the Hailaer River, the water potential changes greatly, and the annual runoff difference is noticeable. In wet years, the water volume of the mainstream flow was so large, the river overflowed onto the wetland on the bank and the river channels increased, resulting in a substantial reduction of vegetation area. However, the adequate water supply improved vegetation growth conditions [49]. In dry years, the mainstream flow became narrow, the oxbow lakes scattered among the wetlands dried up, and the original zigzag tributaries were replaced by new vegetation.

This changed the spatial pattern of the floodplain wetland observed in the image. For the meadow ecosystem, runoff is an important factor affecting NDVI. When runoff was too low, the growth of meadow vegetation deteriorated. The study also found that when the mean NDVI values were calculated without excluding values < 0, there were abnormal cases where the meadow's NDVI was higher than the floodplain wetland's NDVI in several periods of images. We explored the reasons for this effect using the unequal variance t-test, and determined that runoff was the main factor. The runoff of meadow systems was larger than that of wetland systems in the period when NDVI was higher. On the one hand, the increase of water area in the floodplain wetland decreased the mean NDVI; on the other hand, the full water supply improved the growth of the meadow, which also confirmed the significant influence of runoff on the meadow. The sand ribbon is far from the river, and the water retention capacity of soil and plant roots is poor [22], so the effect of runoff is far less than that of precipitation.

The study explored the effect of various environmental factors on NDVI in a statistically significant way. Rather than the actual effect on vegetation, this study focused on the effect on the value of NDVI, aiming to investigate whether short-term environmental factor changes affect NDVI. The results demonstrated that NDVI varies with the humidity and temperature of the day. Therefore, the analysis of NDVI only for a single period is contingent to a certain extent. The comparison of daily NDVI levels in different periods is also uncertain, requiring a comprehensive analysis combined with environmental factors.

## 5. Conclusions

This study calculated the NDVI of 70 Landsat remote sensing images and investigated three types of ecosystems—wetland plain, meadow, and sand ribbon—that co-existed in a small area of Hulunbuir Grassland to analyze the spatial differences, recognize the influencing factors, and further explore the dominant driving factors of each ecosystem and their impact on NDVI. The main conclusions are:

- There were significant differences in NDVI among the three ecosystems, showing wetland plain NDVI > meadow NDVI > sand ribbon NDVI. The data distribution of floodplain wetland conformed to the normal distribution, indicating that vegetation was in a state of natural growth, while meadow and sand ribbon were greatly affected by external interference, presenting a skewed distribution. The multi-year maximum value composite showed a spatial differentiation in that most NDVI values were greater than 0.5 in the floodplain wetland, 0.3–0.5 in the meadow, and less than 0.3 in the sand ribbon ecosystems. The spatial distribution of NDVI was similar to that of altitude.
- The synergistic dominant driving factors of NDVI at the significant level were phenological period, mean relative humidity, average temperature, accumulated precipitation in the first seven days, runoff, and amount of evaporation, which explained 68.8% of the variation of NDVI. The common factors among the three systems were phenological period, precipitation, and humidity. The personalized difference was shown in temperature, runoff, and the response to precipitation aging. The temperature of the floodplain wetland was relatively high, the recharge effect of runoff on the meadow was more remarkable, and the sand ribbon had a significant immediate response to precipitation.
- Among the six dominant factors, phenological period and relative humidity had a significant influence on NDVI and were positively correlated with the three systems. Runoff had little influence, and there was no clear pattern in the data. The responses of temperature, precipitation, and evaporation to the extreme value were strong. The greater the precipitation, the more the NDVI values increased. Temperature and evaporation both increased NDVI within a certain range, but beyond a certain threshold, the opposite effect may occur, influenced by a combination of other factors.

The study expanded on previous research examples of NDVI and its influencing factors, as well as the idea of quantitatively investigating the multi-factor coupling explanation

variation. The study showed it is more convincing to analyze the differences between three ecosystems under the same climate conditions on a statistical level. Different from other research, this study focused on the analysis of the influencing factors and explanatory power of NDVI, rather than the exploration of temporal and spatial changes. However, as the basic data included five phenological periods of vegetation and the growth span was long-term, the correlation coefficient of some factors was not high although the correlation was significant. This study will serve as the basis for follow-up research in which the NDVI evaluation method using multi-phase data superposition is carried out and dynamic changes in the ecosystems are analyzed. This study provides a reference for those involved in the maintenance and protection of natural ecosystems. In future research, data accuracy can also be improved to carry out a comprehensive exploration of NDVI.

**Author Contributions:** Conceptualization, methodology, writing—review and editing, funding acquisition, supervision, C.H.; methodology, software, visualization, writing—original draft preparation, Y.Z.; resources, investigation, validation, X.D.; supervision, J.L. All authors have read and agreed to the published version of the manuscript.

**Funding:** This research was funded by the National Natural Science Foundation of China, grant number U2102209.

**Institutional Review Board Statement:** Not applicable.

**Informed Consent Statement:** Not applicable.

**Data Availability Statement:** The data presented in this study are available on request from the corresponding author. The data are not publicly available due to privacy restriction.

**Conflicts of Interest:** The authors declare no conflict of interest.

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
