# Peer review of "NDVI Characteristics and Influencing Factors of Typical Ecosystems in the Semi-Arid Region of Northern China: A Case Study of the Hulunbuir Grassland"

_land, doi:10.3390/land12030713_

Round 1
Reviewer 1 Report (Previous Reviewer 1)
The manuscript entitled “NDVI characteristics and influencing factors of typical ecosystems in the semi-arid region of northern China: A case study of Hulunbuir Grassland” and authored by Zhao et al., definitely have merits and novelty but need major revision especially English language editing and writing style.
Line 27 - 29 the starting of the statement is not appropriate, need to be re-writ
Line 40 Due to its…????
Line 51-52, the meaning of statement is not clear. Kindly justify what author want to say and rewite the statment
Line 54-57 does these finding belongs to the Lin et al. [15]. kindly justify
Line 60-77 the writing style is very rudendary. This the author need to improve the writing style. Specifically, there is no need to write the objective of study just write the points which need to be related to study and help author to justify the objective of there study
Line 77-79 the authors want to state that low resolution is one of the constraint, still author used to Landsat dataset. When another high resolution datas are availables. Kindly justify
Line 85-87, the structure formation of sentence is not good at all. So kindly re-write
The information regarding the research gaps are missing in the manuscript. Author attempt to add the information, however the representation of the information is not at all good and need addition of further information and restructuring the statements
Line 89 "single period images" kindly justif
Line 92 kindly delete "following" wor
Line 92 "How are" ????
Line 96-100 should belong to material and method section
Overall the introduction part need significant improvement especially the writing style and English language with discussion regarding previous studies
Line 106-107 kindly write proper code for the koppen classification code
Line 122 kindly also write the authority of the species indicated
Line 125 similar to above comment
Line 133 which DEM product was used, kindly mention. Moreover also indicate which Landsat product used for a particular period
Line 147 mention the distance
Line 154 elaborate phenological period what does A to E signifies
Line 212 "70 images of non fixed period" kindly justify. As you have mentioned in the introduction as well as the material method section that the same period images were taken
Line 215 to 216 looking like the author also discussing the results which should comes under Discussion section
Line 220 to 221, the statement "Only in periods with appropriate environmental conditions will 220 they show high NDVI levels." Hard to understand. Kindly re-write
Line 225 to 227 these statements should belong to the material method section rather than result section
Line 246 to 247 the statement will be better suitable in the discussion section
Figure 4. Kindly write the full form of MVC.
Line 252, the detail of the selected "fourteen environmental factors" should be provided in the material method section with additional detail
Line 267-284, the author should strictly stick to the results rather than discussing the results for which you have a separate part
Author need to specify, what are the phenological stages from A to E
Line 316 any relevant source
Line 318 - 319 instead of giving these detail in the text, it should be represented in the figure on both axis
Line 318 -335 the result in this section need to be concise and should be restricted to major findings
Figure 6 and 7 need to revised and should be combined. Authors should adopt some other format to represent the results
Figure 8 the axis should be defined in the figure
Line 414-440 there is repetion of the information
Line 427 The statement should be more relevant to introduction section
Line 435 the scientific name of reed should be provide with authorit
Line 447 -450 the writing style and English language used is not good at all.
Line 476- 491 hardly see any supporting reference
Line 506-536 the authors had made there own argument rather, in scientific research, the authors may have arguments but need to be supported.
Line 526-529 the statement is not clear
Overall the discussion part is very poorly written and in this section the results should be supported with cause and effect rather than only discussing previous studies result.
The Conclusion Part is well written.
Author Response
Dear reviewer,
Thank you for your review. Please see the attachment.

Reviewer 2 Report (Previous Reviewer 2)
Dear authors.
I studied the resubmitted version of your work along with the response to the reviewer's comments. The research questions stated in the introduction are adequately answered. You improved this version and I believe it deserves publication after minor revision considering the following comments:
1. Please, check the text for mistakes in English.
2. Provide higher resolution images for Figure 6 and Figure 7. These figures look blur.
Kind regards
Author Response
Dear reviewer,
Thank you for your review. Please see the attachment.

Reviewer 3 Report (New Reviewer)
The authors take three typical ecosystems of Hulunbeier Grassland - floodplain wetland, meadow and sandy land as the research area. In this paper, 70 Landsat remote sensing images are selected for NDVI monitoring in small-scale and long-term series. They selected environmental variables such as precipitation, temperature and runoff from the factors that may affect NDVI for redundancy analysis, aiming to explore the leading driving factors and their impacts.The research is an interesting research topic, and it also has some practical significance. Overall, the structure of the paper respects the journal's format. I can see that the authors gave a significant effort to make the paper well written and compelling. I think this paper can be published once authors make revisions.
1. The abstract is not concise enough to highlight the innovation of research. Further streamlining is needed to enhance the value of the summary. (lines 13-29)
2.The introduction is not rich enough and must be supplemented. I suggest the author make some necessary additions. The description in this part should directly discuss relevant studies on NDVI characteristics and influencing factorsof typical ecosystems.
3. I highly recommend modifying and update the first sentence of the introduction with the given studies [1,2] as "It is helpful to understand the dynamic changes of ecosystems more deeply and realize the sustainable development of regional ecological environments by systematically monitorring the long-term changes in vegetation and comprehensively analyzing the changing characteristics and driving factors[1,2]."
[1].Application of the Optimal Parameter Geographic Detector Model in the Identification of Influencing Factors of Ecological Quality in Guangzhou, China
[2].The mark of vegetation change on Earth’s surface energy balance
** Authors must have to update the above sentence with provided studies.
4. The figures format of the article is incorrect, please modify it according the requirement of the “Land”.
5. Please update some references from the last five years in the article. Especially the latest achievements in machine learning and intelligence.
6.Write the suggestions at the end of the conclusion.
Round 2
Reviewer 1 Report (Previous Reviewer 1)
The authors have made considerable revisions in the manuscript as per the comments specified and thus be considered for publication in the current form.
Author Response
Dear Professor,
Thank you very much for your careful reading of our manuscript and valuable comments and suggestions. Our manuscript has been greatly improved with your help.
Sincerely yours,
Yating Zhao
Reviewer 3 Report (New Reviewer)
The article has been carefully revised according to the reviewers' suggestions, and the overall has been greatly improved, especially in the introduction where the authors have added some new research results on NDVI. However, I think the abstract section could be further refined to highlight the results of the study. In addition, in the introduction (lines 41-45) "NDVI is useful to help understand the dynamic changes of ecosystems and the surface energy balance more deeply, allowing the sustainable development of regional ecological environments and social development through the systematic monitoring of long-term changes in vegetation [1-2]." Some additional references that can support the conclusion are needed to reflect the scientific integrity of the article, which will be more beneficial for the dissemination of the article.
[1]Demuzere, M., Orru, K., Heidrich, O., Olazabal, E., Geneletti, D., Orru, H., Bhave, A.G., Mittal, N., Feliu, E., Faehnle, M., 2014. Mitigating and adapting to climate change: Multi-functional and multi-scale assessment of green urban infrastructure. J. Environ. Manage. 146, 107–115. https://doi.org/10.1016/j.jenvman.2014.07.025
[2]Local climate zone ventilation and urban land surface temperatures: Towards a performance-based and wind-sensitive planning proposal in megacities. Sustain. Cities Soc. 47, 101487. https://doi.org/10.1016/j.scs.2019.101487
It's best to replace the images in the original article with high resolution ones.
Good luck!
Author Response
Dear Professor,
Thank you very much for your careful reading of our manuscript and your valuable comments and suggestions. We have revised our manuscript according to the comments. The detailed response to the comments is as follows:
1) The abstract section has been further refined and revised. (Lines 16–22)
2) The introduction has been supported by the references to reflect the scientific integrity of the article. (Lines 44)
3) The resolution of figures has improved. (Figures 1,4,7,8)
Besides the above changes, we have corrected some errors.
If you need any other information, please don't hesitate to contact me.
Yours sincerely,
Yating Zhao
This manuscript is a resubmission of an earlier submission. The following is a list of the peer review reports and author responses from that submission.
Round 1
Reviewer 1 Report
In the manuscript, 'NDVI characteristics and influencing factors of typical ecosystems in the semi-arid region of northern China: A case study of Hulunbuir Grassland' investigated by Zhao et al.,. The authors put great effort into conducting the study however, I have some minor concerns with it mentioned in the attached file, especially in the discussion and conclusion section

Reviewer 2 Report
Dear authors, your research is interesting but in my opinion you have overestimated the results of your analysis based on Landsat images and metereological data from stations in the study area. Based on the following comments I propose rejection at this stage and resubmission after major corrections.
1. What is the innovation of this study compared to similar studies from all over the world?
2. Why you did not used some higher resolution satellite images or even drone images to check the accuracy of the Landsat images? At least for a part of the study area! The 30 m spatial resolution of the Landsat images is low and this is possibly the reason for the low correlation of the NDVI with most of the selected factors. Please check the bibliography for similar published works.
3. Table 1. Most of the correlated factors, except the phenological period, present medium to low correlation with the NDVI. This makes your results unstable.
4. Discussion: You present the interpretation of your results with nο adequate comparisons with similar studies from other regions of the world. What is the contribution of this work to the specific field of remote sensing?
5. References: Very few references compared to the published work in this scientific field.
Kind regards
Reviewer 3 Report
This manuscript aims at NDVI of different ecosystems and their responses to environmental factors. This study demonstrates driving effect in the short term on a statistical level, and provided a scientific basis for future NDVI research. The paper is well written and easy to understand, and the statistics used are appropriate.
Abstract
The novelty of the research needs to be supplemented
Results
Add the explanation first axis and the second axis in Fig.5
Discussion
The discussion generally needs improvement. Considering the regional differences of driving factors, the interaction of soil conditions and climate affects the vegetation characteristics of ecosystem. e.g. “Soil development mediates precipitation control on plant productivity and diversity in alpine grasslands”